# Doppler power spectrum processing methods for a vertical sensing 94 GHz millimeter-wavelength cloud radar

Hai Lin<sup>1,2</sup>, Jie Wang<sup>1,2</sup>, Zhenhua Chen<sup>1,2,3</sup>, and Junxiang Ge<sup>1,2,3</sup>

<sup>1</sup>School of Electronic and Information Engineering, Nanjing University of Information Science & Technology, Jiangsu 210044, China
 <sup>2</sup>Institute of Electronics Information Technology and System, Nanjing University of Information Science and Technology, Nanjing 210044, China
 <sup>3</sup>Jiangsu Key Laboratory of Meteorological Observation and Information Processing, Jiangsu 210044, China
 Correspondence: Junxiang Ge (jxge@nuist.edu.cn)

**Abstract.** A set of Doppler power spectrum processing methods for TJ-II, a vertical sensing 94 GHz millimeter-wave cloud radar(MMCR), is proposed to distinguish clouds and enhance the data quality. The noise level is estimated by a modified segment method with 2-D segments. A two-step cloud signal mask method is proposed to distinguish clouds and noise. A Gaussian filter with adaptive standard variance is used to improve detection performance at the boundary of clouds and noise. Square

signal blocks were constructed to test our method. Velocity dealiasing is carried out combined with a pre-dealiasing processing and the dual Pulse Repetition Frequency (PRF) technique to address a particular phenomenon called half-folded in MMCRs, which can not be fixed by post-processing at the base datum stage. Some observations of TJ-II are used to demonstrate our method. A comparison between our method as pre-processing and the method as post-processing proposed by Key Laboratory for Semi-Arid Climate Change of the Ministry of Education and College of Atmospheric Sciences, Lanzhou University, was

carried out using one-day observation. It was found that our method shows some advantages in cloud base detection.

#### 1 Introduction

Clouds cover almost two-thirds of the global surface. They are composed of water droplets, supercooled water droplets, ice crystals, and their mixtures condensed from water vapor in the atmosphere. Clouds contribute to our lives in both direct and indirect ways. Clouds are the most visible elements of the sky and the dominant contributors to the weather we experience

every day. Clouds play an important role in climate change, global radiation budget, and weather forecasting(Muller and Fischer, 2007; Rosenfeld, 2006; Arking, 1991). Clouds control Earth's weather and regulate its climate(Solomon et al., 2007; Collins et al., 2007). Clouds also play a critical role in the water cycle and shape the global distribution of water resources.

Despite the importance of clouds in the climate system, they are difficult to represent in climate models(Bony et al., 2015). Clouds constitute the largest single source of uncertainty in climate prediction(Baker and Peter, 2008). Therefore, the study of clouds is integral to meteorological and climate research.

Cloud researches are complex and huge, from macroscopic cloud amount, cloud shape, cloud height, and cloud velocity detection to microscopic cloud particle physics, chemistry, optics, radiation characteristics, etc. Cloud sensing is the founda-

55

tion of cloud research. Clouds have been the subject of observation for centuries, but serious systematic investigations began only a few decades ago(Lamb and Verlinde, 2011). The current common remote sensing devices for clouds include weather

- radar, cloud lidar, and millimeter-wave cloud radar(MMCR)(Ran et al., 2021; McGill et al., 2002; Lhermitte, 1987a). These devices have different application scenarios. Weather radar mainly operates at S-band and C-band, such as WSR-88D, CIN-RAD/CC, with low atmospheric propagation loss and extremely high peak transmit power, making it the most indispensable tool for remote sensing of large-scale clouds and precipitation(Kumjian, 2018). Lidar is capable for boundary layer, Cloud-Top Height(CTH), Cloud Base Height(CBH) and aerosol sensing with high resolution(Ansmann and Müller, 2005; Baars et al.,
- 2008; Müller et al., 2007; Kim et al., 2011; Stephens et al., 2002). On the opposite, Lidar is relatively poor at detecting thick clouds due to poor penetration.

Millimeter-wave radars(MMCR) are somewhere in between weather radar and lidar, with higher spatial and temporal resolution than weather radars and better penetration than lidar, making it the better tool for cloud vertical structure(CVS) sensing(Yan et al., 2021). Not only the macro parameters such as cloud thickness, cloud-top height, cloud-bottom height, and cloud amount

but also the microphysics parameter such as cloud particle size, concentration, droplet spectrum distribution, etc. could be obtained by MMCRs(Kropfli et al., 1995; Kollias et al., 2007; Verlinde et al., 2013).
 Because of its outstanding advantages for cloud research, MMCR has been deployed on various research platforms, including

Cloud-Sat with Cloud Profiling Radar(CPR, Atmospheric Radiation Program (ARM) with Ka-band Zenith Radar(KAZR) .etc(Im et al., 2005; Chandra et al., 2015). The 94 GHz(W-band) and 35 GHz(Ka-band) are the common frequencies MMCRs

- work at. Compared with the 35 GHz, 94 GHz MMCR has a shorter wavelength with a stronger cloud particle backscatter, leading to better detection performance. 94 GHz MMCR was first developed by Lhermitte (1987a, b). Nowdays, there are a variety of 94 GHz MMCR(Takano et al., 2012; Huggard et al., 2008; Danne et al., 1999; Wu et al., 2014; Hogan et al., 2003; Delanoë et al., 2016). They almost take one of the two systems. One is Pulse Doppler(PD) radar with a Traveling Wave Tube Amplifier(TWTA) and one or two antennas. The other is Frequency Modulated Continuous Wave(FMCW) radar with a Solid-
- State Power Amplifier(SSPA) and two antennas. The PD MMCR always used TWTA due to the insufficient peak power of W-band SSPA in the past, while FMCW MMCR always used two antennas due to insufficient isolation of the W-band circulator. The PD MMCR with TWTA suffers from power supply requirements and lifetime. FMCW MMCR with two antennas suffers in size and weight. Benefiting from the rise in power of the 94G SSPA in recent years, to address the deficiencies of these two kinds of radars due to size, power demand, and reliability, Our team developed a PD MMCR called TianJian II(TJ-II) with
- a single antenna and SSPA. Naturally, a set of Doppler power spectrum processing methods matching TJ-II characteristics is proposed.

The TJ-II MMCR is described in Sect.2. A noise level estimating method based on the segment method is described in Sect.3. A cloud signal mask method with two-step is described in Sect.4. A velocity dealiasing method combined with the dual pulse repetition frequency (PRF) technique is described in Sect.5. Examples of our method applied to TJ-II observations are shown in Sect.6

Figure 1. Photo of TJ-II MMCR

# 2 The TJ-II radar

TJ-II cloud radar, developed by Nanjing University of Information Science & Technology(NUIST), is a groud-base vertical detection cloud radar. It works at about 94 GHz for dual-polarization measurements. It is a PD radar using SSPA and has only a single antenna with two polarized ports(Wang et al., 2022). Segment detection and pulse compression technology are used
to overcome the less output power of SSPA than TWTA. Also, the dual PRF technique is used for velocity dealiasing with the dedicated algorithm. The primary purpose of the TJ-II is to provide a small mass and size, and low-cost 94 GHz MMCR. It can provide reflectivity, mean Doppler velocity and spectral width, and linear depolarization ratio(LDR). The performance metrics of the TJ-II are shown in Tab.1. The peak power of the transmitter is 6 W. The pulse length differs from 1 us to 20 us according to the detection altitude. The detection altitude ranges from 300 m to 15 km, covered by three pulse lengths with an 18.75 m range gate. The pulse repetition time is between 100us and 150us to balance the unambiguous range and Nyquist velocity.

# 3 Noise level estimation

Similar to other PD radars, after pulse compression and slow-time FFT processing, we can get the Doppler power spectrum of the cloud signal. The Doppler power spectrum reflects the power distribution corresponding to particles with different Doppler velocities. Power spectrum data is related to cloud microphysics and dynamics, which can be obtained by some inversion

methods. Also, the Doppler power spectrum is the base of reflectivity, mean Doppler velocity, and spectral width. So power spectrum processing method will directly affect the data quality based on it.

The first is to estimate the noise level. Noise level is the mean noise power in the power spectrum, which is essential for distinguishing signal and noise. The accuracy of the noise level will directly affect the performance of cloud signal detection

90

#### Table 1. Performance metrics of the TJ-II MMCR

| Frequency              | 94 GHz                      |  |  |
|------------------------|-----------------------------|--|--|
| Transmitter type       | Solid-state power amplifier |  |  |
| Peak transmitter power | 6 W                         |  |  |
| Antenna type           | Singal Cassegrain antenna   |  |  |
| Antenna gain           | 50.8 dBi                    |  |  |
| Antenna beamwidth      | $0.45^{\circ}$              |  |  |
| Vertical sampling      | 18.5 m                      |  |  |
| Pulse width            | 1.5/5/20 us                 |  |  |
| Pulse repetition time  | 100/150 us                  |  |  |
| FFT points             | 128/512                     |  |  |
| Sensitivity            | -30 dBZ                     |  |  |
|                        |                             |  |  |

and then affect the calculation of the spectral moment. In the early days, the noise level was set to a fixed value(Battan,

- 1964; Sekhon and Srivastava, 1971). However, the power spectrum will change, and the performance will be poor in different atmospheric backgrounds. In wind profile radar, it is common to treat the mean of the farthest range gate spectrum as the noise level, assuming there is only noise and no signal(Feng et al., 2021). However, in our application, the maximum detection range of TJ-II is 15 km, and some kinds of high clouds exist at this height. So this method is not suitable for TJ-II. Hildebrand and Sekhon (1974) proposed an objective determination method for noise level based on the physical properties of white noise.
- This method was wildly used in many weather radars. Objective determination starts from the high-power position, searches downwards, and gradually strips the cloud signal until the signal and noise are entirely separated. This method is based on rigorous theory. But in practical application, the amount of calculation is significant, and it is not easy to meet the judgment condition of loop termination. Liu et al. (2014) proposed a method assuming that there is only noise in the power spectrum at a velocity over 8  $ms^{-1}$ , and the mean of these points is the estimated noise level. This method was applied on a Ka-band
- MMCR with a Nyquist velocity of over 8  $ms^{-1}$ . When it comes to W-band MMCRs, the Nyquist velocity is about one-third of Ka MMCRs with the same Pulse Repetition Time (PRT). For example, in TJ-II, the maximum detection range is 15 km, and the corresponding minimum PRT is 100 us, leading to only about 8  $ms^{-1}$  Nyquist velocity. So LIU's method is incapable of TJ-II because of the narrow Nyquist velocity interval. The velocity dealiasing processing will be discussed in Sect5. Petitididier et al. (1997) proposed a segment method that divides the power spectrum into multi segments and chooses the minimum mean

of these segments as the noise level based on the assumption that cloud signals will not fill the entire power spectrum.

The noise level estimation method in TJ-II is based on the segment method. The following shows the principle of the segment method.

According to the central limit theorem(CLT), the properly normalized sum of independent random variables tends toward a normal distribution. The mean of N random samples with overall mean  $\mu$  and variance  $\sigma^2$  follows a normal distribution  $N(\mu, \sigma^2/N)$ . Meanwhile, the cumulative distribution function(CDF) of the minimum value of M random variables X can be