# Peer review of "Doppler power spectrum processing methods for a vertical sensing 94 GHz millimeter-wavelength cloud radar"

_Atmospheric Measurement Techniques, 2022_

## Author Comment (AC1)

**Response to Anonymous Referee #1**

We thank the reviewer for his/her constructive comments and suggestions on this manuscript, which are very helpful for us to improve our paper. Our responses to these comments are given below.

*- The lengthy introduction section covers irrelevant details of general cloud research but contains absolutely nothing about the actual problem addressed in the manuscript. The introduction must include a general overview of problems in the data processing, literature review on this topic, and explain which of these problems are solved in the study. In the current status, the motivation of the manuscript is not defined.*

**Response:**
Thank you for pointing out the deficiencies in our introduction writing. The main motivation of this manuscript is as follows:

The purpose of our design of TJ-II radar was to develop a cloud radar that is easy to promote and deploy. This puts a series of requirements on the radar, including economic cost, performance, usability, etc. The economic cost is mainly determined by the system hardware. The performance and usability are determined by both hardware and software. For performance, since TJ-II MMCR is a single antenna pulse Doppler radar with SSPA, it has less mean transmit power than both traditional pulse Doppler radars with TWTA and FMCW radars with two antennas. More weak signal situations will be encountered in detection, which needs more care in signal processing. For usability, our algorithm needs to be able to deal with various situations.

As a result, in the development of TJ-II, we found that existing methods have some limitations in dealing with different situations. Therefore, we modified these methods to adapt to more situations and improve the performance of weak signal detection. All the responses below are based on this motivation to explain why we don't use the methods mentioned in the comment.

We're going to overhaul the introduction to better show our motivation.

*- The manuscript describes already existing methods slightly fine-tuned for the presented radar system. In my opinion, there is a severe lack of novelty in the manuscript. (1) In the section 3 the authors apply the segment method developed decades ago, the only difference is that the method is applied not just to a spectrum in a single range bin but to several range bins. It is hard to consider this as a novel approach. Also, it is not clear why the authors try to solve the noise estimation topic at all, since with modern computers there is absolutely no problem in applying the Hildebrand-Sekhon algorithm in real time.*

**Response:**
Aside from the computational load of the Hildebrand-Sekhon algorithm, there are some reasons we didn't take advantage of it:

(1) The estimated noise level of range gates with signals will be significantly higher than the non-signal area. As shown in Fig.1, the difference between estimated noise levels is over 3dB. Since the detection of weak signals is more sensitive to noise level, the deviation fluctuation of the estimated noise level will greatly affect the detection performance of weak signals. As shown in Fig.2, just 1dB higher estimated noise level has greatly deteriorated the detection of weak signals.

(2) As mentioned in Hildebrand-Sekhon 1974, the wider the width of the signal spectrum (velocity width) compared with the total frequency span (Nyquist velocity), the worse the estimated noise level due to the reduction in the number of points used for noise estimation. Our TJ-II radar, working at 94GHz, has less Nyquist velocity, leading to a wider proportion with the same signal velocity width than other lower frequency radars. It reduced the usability of the Hildebrand-Sekhon algorithm in more conditions.

In general, our requirement for noise level estimation is more accurate and stable for more situations. More samples lead to more accurate estimates. So, we expand the segments from one range gate to several range gates. Also, we select partial segments from the entire spectrum, not dividing the entire spectrum to reduce computation, nor from a specific area like farthest range gates or maximum velocity areas to prevent the presence of signals in these areas in some situations.

[Figure]

Fig.1 Comparison of noise level estimation methods (a) Raw power spectrum (b) Noise level estimated by three methods

[Figure]

Fig.2 Effect of noise level estimation bias on weak signals. (a) Raw power spectrum (b) cloud signal mask result with noise level estimated by our method (c) cloud signal mask result with noise level 1dB higher than (b)

*- (2) In the section 4, the authors suggest using a time-space filter to improve sensitivity. Such filters, however, have been routinely applied for decades (e.g. Clothiaux et al. 1995 JAOT, Marchand et al. 2008 JAOT). The performance of the authors new filter is very close to the Gaussian one. For a single presented case study (which is hard to consider as statistically significant), FAR and MDR are nearly the same (just a few % difference).*

**Response:**

Indeed, the method of using time-space filters is not new, but the core lies in the filter kernels. The filter used in reference was designed based on the assumption that the noise obeys a normal distribution and for post-processing. At the power spectrum stage, the noise spectrum lines obey an exponential distribution according to our observation. At the post-processing stage, the noise is the sum of noise spectrum lines within one range gate and obeys normal distribution according to the central limit theorem. Therefore, it does not fit our use case in the power spectrum stage. Also, the referenced filters need at least five iterations. It needs a large computational source for real-time processing of the power spectrum stage. If combined with the Hildebrand-Sekhon algorithm for noise level estimating, there are fewer computing resources for the other processing of the whole system.

From the perspective of absolute value, it does not improve much and the overall error rate (mean of FAR and MDR) of our filter and the Gaussian filter at offset 0 is very similar (26.12% vs 25.75%). But from terms of relative value, at offset 1,2 they are (3.56% vs 4.70%) and (1.35% vs 3.10%), improved by 24% and 56% respectively. Another key performance is MDR in the full signal area. It is improved by 75% for weak signals.

*- (3) In the section 5 the aliasing problem is solved by applying the dual pulse repetition frequency technique which has also been used for decades. Authors themselves give a long list of references. The spectral dealiasing is also not a new thing (e.g. Kuehler et al. 2017, Maahn and Kollias 2012).*

**Response:**

The dual pulse repetition frequency technique is indeed a common technique, especially for common radars. But when it comes to the cloud radar, the targets, cloud, become body targets with a larger spectral width. It may cause a special half-folded state and needs extra pre-processing as described in the manuscript. Otherwise, a wrong result will be obtained. The special half-folded state was not mentioned in the references about the dual pulse repetition frequency technique. We also improved the selection method of positive and negative partitions to adapt to more situations.

There are some spectral dealiasing methods with single pulse repetition frequency, as mentioned in the comment, Maahn and Kollias 2012 and Zheng et al. 2016. They both used the iterative method. The difference between their methods is the determination of the initial value (or trusted value). Maahn and Kollias used the empirical relationship between the reflectivity factor and velocity. Zheng used the assumption that there is no aliasing at the cloud top. Also, the iterative process is also based on the condition that the velocity of adjacent range gates is continuous (the difference is not large). When any one of the initial conditions or iteration conditions fails, all subsequent results will fail. For example, the empirical relationship will change in different areas and weather conditions. Manual corrections are required for radar deployment and use. Continuity of velocity may fail for strong turbulence. Because of these limitations, we have used the dual pulse repetition frequency technique which even needs the cooperation of hardware. It will only fail due to the limitations of the radar itself, such as velocity over the new Nyquist velocity, or spectral width over the whole spectrum. Also, the result of each range gate is independent, one failure will not affect others.

Overall, all our methods were proposed to adapt to more situations and weak signal detection, improving the performance and usability of the TJ-II.

**Reference**

Clothiaux, E., Miller, M., Albrecht, B., Ackerman, T., Verlinde, J., Babb, D., Peters, R., and Syrett, W.: An evaluation of a 94-GHz radar for remote sensing of cloud properties, Journal of Atmospheric and Oceanic Technology, 12, 201–229, 1995.

Hildebrand, P. H. and Sekhon, R.: Objective determination of the noise level in Doppler spectra, Journal of Applied Meteorology (1962-1982), pp. 808–811, 1974.

Maahn, M. and Kollias, P.: Improved Micro Rain Radar snow measurements using Doppler spectra post-processing, Atmospheric Measurement Techniques, 5, 2661–2673, 2012.

Marchand, R., Mace, G. G., Ackerman, T., and Stephens, G.: Hydrometeor detection using CloudSat—An Earth-orbiting 94-GHz cloud radar, Journal of Atmospheric and Oceanic Technology, 25, 519–533, 2008.

Zheng, J.-F., Liu, L.-P., Zeng, Z.-M., Xie, X., Wu, J., and Feng, K.: Ka-band millimeter wave cloud radar data quality control, J. Infrared
Millim. Waves, 35, 748–757, 2016.

---

## Author Comment (AC2)

**Response to Anonymous Referee #2**

We thank the reviewer for his/her constructive comments and suggestions on this manuscript, which are very helpful for us to improve our paper. Our responses to these comments are given below.

*-This manuscript presents a step by step procedure to process Doppler spectrum data collected by a millimeter wavelength weather radar. The presentation is clear and detailed but it is unclear what is new in the proposed technique.*

**Response:**
Thank you for pointing out the deficiencies in our writing. The main improvements compared with conventional methods include 2-D segment method for noise level estimation, a new adaptive filter based on Gaussian kernel for cloud signal detection, and half-folded dealiasing processing before conventional dual PRF processing.

*-In the first step of the processing for example the noise floor is estimated using the standard segment method, but extended to multiple range gates. Seems like an insufficient contribution to prior publications. Also, addressing such a mature topic should include a comparison of other techniques based on the quality of results. The well known Hildebrand and Sekhon (1974) (H-S) noise estimation technique for example is mentioned to require significant calculations. This may have been a deterent for real-time processing in 1974, but since then computers have significantly improved so a comparison using real and simulated data of the proposed 2-D segment method would be interesting. If the H-S method is still computationally too demanding, then let's show that.*

**Response:**
The 2-D segment method comes from our demand on the low bias estimated noise level, which is directly related to the number of points used for noise estimation. Because our pulse Doppler system with SSPA, weak cloud signal detection must pay attention. As shown in Fig.1, even 1 dB higher estimation will have a great impact on weak signal detection.

[Figure]

Fig.1 Effect of noise level estimation bias on weak signals. (a) Raw power spectrum (b) cloud signal mask result with noise level estimated by our method (c) cloud signal mask result with noise level 1dB higher than (b)

The well-known Hildebrand and Sekhon(H-S) method did be evaluated. The computational demand of H-S method relays on the optimization of program, including ranking and calculation algorithm. A basic demo of H-S method we did in MATLAB consumed about 33 ms. The parameters for simulation were 512 points per range gate with 128 points signal, and 800 range gates. As a comparison, 2-D segment method only took 3 us. Also, even the next cloud signal detection took only about 10 ms. The average frame time is 64 ms (125 us per pulse and 512 pulses). The H-S method almost took over half of frame time.

A much high-end processor or algorithm optimization can decrease the time, but computational demand is not the key. The key is that the stability of estimated noise level by H-S method don't meet our requirement. The strength of cloud signal will affect the bias of estimated noise level, as shown in Fig.2 and Fig.3. Strong cloud signal works fine while Weak signal leads to higher estimation. However, it is also weak signals that are more susceptible to noise bias as shown in Fig.1. Fig.4 shows an actual example of our observation data and confirms this phenomenon. Therefore, we did not use the H-S method.

[Figure]

Fig.2 Simulation of H-S method with 512 points per range gate with 128 points signal with different peak SNR. (a1) is the spectral lines for simulation with peak SNR 6 dB. (a2) is the noise level estimated by H-S method. (b) is the same as (a) with peak SNR 20 dB.

[Figure]

Fig.3 Estimated noise level by H-S method with different peak SNR

[Figure]

Fig.4 Example of one TJ-II observation data (a) Raw power spectrum (b) Noise level estimated by three methods.

*-It is also not clearly stated if any new technique is proposed in the second step of identifying the cloud signals. It seems that this processing sequence follows the two references provided, yet it is presented with great detail like this was a novel method.*

**Response:**
In the step of cloud signal detection, an adaptive Gaussian filter based on deviation of the mean value in the filter window and threshold.

$$\sigma_g = \sigma_0 \cdot \max \left( \frac{T_s}{\mu(x,y)}, \frac{\mu(x,y)}{Ts} \right)$$

The main idea is to use the statistics of the window area to judge the trend of edge side and then change the weighting ($\sigma_g$) to enhance the trend.

During open discussion, another adaptive Gaussian filter combined with Kuwahara filter was proposed based on the same idea. Here is the description of the new improved filter:

1 Divide the window $(2k + 1)$ into four subregions (k+1) as the Kuwahara filter do, as shown in Fig.5

| a | a | a/b | b | b |
|---|---|-----|---|---|
| a | a | a/b | b | b |
| a/c | a/c | a/b /c/d | b/d | b/d |
| c | c | c/d | d | d |
| c | c | c/d | d | d |

Fig.5 Subregion division

2 Calculate the ratio of mean $m_i$ and standard deviation $\sigma_i$ of the four subregions.

$$r_i = \frac{m_i}{\sigma_i}$$

3 Generate four Gaussian filters $(2k + 1)$ with four standard deviations $\sigma_{gi}$ according to

the ratio $r_i$, and normalize them with their center pixel.

$$\sigma_{gi} = (\frac{r_i}{0.88})^2 \sigma_0$$

4 Fill the four subregions of final filter with the corresponding subregions of corresponding Gaussian filter factors. The overlap areas use the mean value of factors.

5 Normalize the final filter and apply to the original window.

The ratio of mean and standard deviation of mixed signal are shown in Fig.6. Both noise and signal are exponential distribution with rate parameter 1 and 3, respectively. When the mixed signal is purer, the ratio is closer to 1. Based on this feature, we use the ratio to determine the mixing degree of the subregion, the lower the mixing degree (closer to 1), the greater the weighting ratio (larger $\sigma_{gi}$). The factor 0.88 is the mean of 1 and lowest ratio about 0.77. The square is to enhance the trend.

[Figure]

Fig.6 Ratio of mean and standard deviation of mixed signal.

The performance of the new adaptive filter is shown in Table. The simulation parameter is the same as the one in preprint, where noise and signal rate parameters are 1 and 3, respectively and the threshold $T_s$ is 1.8.

Table.1 Simulation performance of FAR and MDR using different filters.

| Filter type | Performance | Offset | | | |
|---|---|---|---|---|---|
| | | 0 | 1 | 2 | 3 |
| Filter in Response | FAR | 30.3% | 2.94% | 0.66% | 0.56% |
| | MDR | 16.96% | 5.67% | 2.55% | 1.30% |
| | Mean | **23.63%** | **4.31%** | **1.60%** | **0.93%** |
| Filter in preprint | FAR | 33.2% | 2.25% | 0.45% | 0.14% |
| | MDR | 19.27% | 4.97% | 2.24% | 1.33% |
| | Mean | 26.24% | 3.61% | 1.35% | 0.74% |
| Gaussian filter | FAR | 29.55% | 2.16% | 1.07% | 1.07% |
| | MDR | 22.04% | 7.26% | 5.27% | 4.97% |
| | Mean | **25.80%** | **4.71%** | **3.17%** | **3.02%** |
| Signal rate parameter 10 | | | | | |
| Filter in Response | Mean | 43.95% | 8.48% | 1.26% | 0.22% |
| Filter in preprint | Mean | 49.99% | 44.41% | 5.29% | 0.03% |
| Gaussian filter | Mean | 48.64% | 11.90% | 0.67% | 0.46% |

Compared with the normal Gaussian filter, the mean failure rate (FAR and MDR) of our new filter is better at all offsets. Another improvement is the stability. We found our previous filter in preprint are less robust when the signal rate parameter increase. As shown

in the bottom of the Table.1, when the signal rate parameter raised to 10, performance of the filter in preprint deteriorates rapidly at offset 1. The improved filter in this response is still better than the Gaussian filter overall.

*-Similarly the velocity dealiasing using dual PRF is not new.*

**Response:**
Velocity dealiasing using dual PRF is indeed quite common. In this preprint, it is not mainly about how to perform dual PRF processing, but about the pre-processing before dual PRF processing. There will be a phenomenon called half-folded, shown in Fig.7(a1), caused by the wide spectral width of cloud signals, which does not exist in general point target radar. Half-folded will result in a wrong initial velocity not the equal to the $V_t \pm 2nV_m$ when calculation the initial velocity. It breaks the principle of dual PRF and results in wrong final velocity. There are few references talk about this phenomenon when applying dual PRF in vertical sensing cloud radar. The preprint proposed the method to detect the half-folded and a simpler and universal method of dividing positive and negative intervals, the median method.

[Figure]

Fig.7 One example of TJ-II observation data with a half-folded state. (a1) and (a2) is the mask result by cloud signal mask processing with PRT 150us and 100us, respectively. (b1) is the raw mean velocity of (a1) with and without pre-defolding processing. (b2) is the final dealiased velocity with and without pre-defolding processing

*-The presentation needs to make a clearer distinction between what's new and when an existing technique is applied - when the reference is sufficient instead of a lot of detail. Overall this is nice work, but it is more like a documentation of the radar than a journal*

*publication advancing the state of science.*

**Response:**
Thanks for your comments, we will improve revision to clear what is new in the proposed technique.